# Active pharmacovigilance of the seasonal trivalent influenza vaccine produced by Instituto Butantan: A prospective cohort study of five target groups

Tazio Vanni[1]☺*, Beatriz da Costa Thomé[2]☺, Mayra Martho Moura de Oliveira[1‡], Vera Lúcia Gattás[1‡], Maria da Graça Salomão[1‡], Marcelo Eiji Koike[1‡], Maria Beatriz Bastos Lucchesi[1‡], Patrícia Emília Braga[1‡], Roberta de Oliveira Piorelli[3‡], Juliana Yukari Koidara Viscondi[1‡], Gabriella Mondini[4‡], Anderson da Silva[1‡], Heloísa Maximo Espínola[3‡], Joane do Prado Santos[3‡], Samanta Hosokawa Dias de Nóvoa Rocha[5‡], Lily Yin Weckx[2‡], Olga Menang[6‡], Muriel Soquet[6‡], Alexander Roberto Precioso[1,7‡]

1 Clinical Safety and Risk Management Centre, Instituto Butantan, São Paulo, São Paulo, Brazil, 2 Federal University of São Paulo, São Paulo, São Paulo, Brazil, 3 Clinical Trials Centre, Instituto Butantan, São Paulo, São Paulo, Brazil, 4 Oxford University, Oxford, United Kingdom, 5 Federal University of Roraima, Boa Vista, Roraima, Brazil, 6 Center for Vaccine Innovation and Access, Program for Appropriate Technology in Health (PATH), Geneva, Switzerland, 7 Pediatrics Department, Child Institute of the Clinics Hospital of the School of Medicine of University of São Paulo, São Paulo, São Paulo, Brazil

☺ These authors contributed equally to this work.
‡ These authors also contributed equally to this work
* tazio.vanni@butantan.gov.br

**Data Availability Statement:** All relevant data are within the paper and its Supporting information files.

## Abstract

### Introduction

Active pharmacovigilance studies are pivotal to better characterize vaccine safety.

### Methods

These are multicenter prospective cohort studies to evaluate the safety of the 2017 and 2018 seasonal trivalent influenza vaccines (TIVs) manufactured by Instituto Butantan, by means of active pharmacovigilance practices. Elderly, children, healthcare workers, pregnant women, and women in the puerperium period were invited to participate in the study during the 2017 and 2018 Brazilian national seasonal influenza vaccination campaigns. Following immunization, participants were observed for 30 minutes and they received a participant card to register adverse events information. All safety information registered were checked at a clinical site visit 14 days after immunization and by a telephone contact 42 days after immunization for unsolicited Adverse Events (AE) and Guillain-Barré Syndrome (GBS).

### Results

A total of 942 volunteers participated in the two studies: 305 elderly, 109 children, 108 pregnant women, 32 women in the postpartum period, and 388 health workers. Overall, the

**Funding:** The study was funded by Instituto Butantan.

**Competing interests:** TV, MGS, JYKV, RPG, MTRPC, ROP, PEB, AS, HME, JPS, VLG, MEK, MBBL, MMMO, SHDNR, LYW and ARP are employees of Instituto Butantan. BCT and GM are former employees of Instituto Butantan. SHDNR and LYW received financial aid for their roles as principal investigators. This does not alter our adherence to PLOS ONE policies on sharing data and materials.

median number of AR per participant ranged from 1 to 4. The lowest median number of AR per participant was observed among healthcare workers (1 AR per participant) and the highest among pregnant women (4 AR per participant). Overall, local pain (46.6%) was the most frequent solicited local AR. The most frequent systemic ARs were: headache (22.5%) followed by fatigue (16.0%), and malaise (11.0%). The majority of solicited ARs (96%) were mild, Grades 1 or 2), only 3% were Grade 3, and 1% was Grade 4. No serious AEs, including Guillain-Barré Syndrome, were reported up to 42 days postvaccination.

## Conclusion

The results from the two studies confirmed that the 2017 and 2018 seasonal trivalent influenza vaccines produced by Instituto Butantan were safe and that active pharmacovigilance studies should be considered, when it is feasible, as an important initiative to monitor vaccine safety in the post-marketing period.

## 1. Introduction

According to the World Health Organization (WHO), influenza infection results in 3 to 5 million severe cases and 250,000 to 500,000 deaths annually [1,2]. The most effective measure to prevent influenza is annual vaccination. The Ministry of Health (MoH) of Brazil, through its National Immunization Program (NIP), has promoted annual national influenza vaccination campaigns since 1999. The annual immunization campaigns target some of the following populations: children between six months and five years of age who receive two doses, 30 days apart when vaccinated for the first time, elderly, pregnant women, women in the puerperium period, health workers and then the general population.

In 1999, a technology transfer partnership between Instituto Butantan (IB) and Pasteur-Mérieux, currently Sanofi Pasteur, was established to produce split virus inactivated trivalent influenza vaccine (TIV) propagated in embryonated eggs [3]. In 2013 IB released the first batches of vaccines entirely produced by the Institute. From 2013 to 2019 approximately 260 million doses of TIV were produced and delivered to the Brazilian MoH.

Pharmacovigilance is the practice of detecting, assessing, understanding, responding, and preventing adverse drug reactions, including reactions to vaccines. It is now an integral part of the regulation of drug and vaccine safety. National post-licensure vaccine safety monitoring systems vary considerably in their structure, methods, and performance, with disparities occurring particularly between high-, middle- and low-income countries [4].

Post-marketing vaccine surveillance for adverse events (AEs) has typically relied on spontaneous reporting. However, regulatory agencies have recently turned their attention to more active approaches emphasizing the importance of active pharmacovigilance [5].

Instituto Butantan has been conducting routine pharmacovigilance activities for its TIV using a passive method of surveillance. As part of the preparedness for the WHO pre-qualification inspection of the seasonal influenza vaccine [6], we conducted active pharmacovigilance studies in 2017 and 2018. Therefore, the objective of those studies were to perform active surveillance of Adverse Events Following Immunization (AEFI) in the following five target groups from the 2017 and 2018 annual influenza immunization campaigns in Brazil: elderly, children, healthcare workers, pregnant women, and women in the post-partum period.

## 2. Material and methods

### Study designs

We performed multicenter observational prospective cohort studies aimed at identifying and evaluating, actively, AEFI among subjects vaccinated (intramuscularly) with the trivalent influenza vaccines [TIV] (split virion, inactivated) propagated in embryonated eggs [3], manufactured by IB in 2017 and 2018, respectively. The studies were conducted at the Centro de Referência de Imunobiológicos Especiais (CRIE) Federal University of São Paulo (UNIFESP) in 2017 and 2018, and at the research site Federal University of Roraima, Boa Vista—Buritis in 2017. The studies were approved by the Ethics Committee of the Federal University of São Paulo (Protocol approvals: 2017, No—2034906; 2018, N° -2517399) and of the Federal University of Roraima (Protocol approval in 2017: N°– 2045362), and were conducted in accordance with the principles of the Declaration of Helsinki, Good Clinical Practices [7].

The two studies were registered with ClinicalTrials.gov: NCT03057483 in 2017; NCT03392207 in 2018, and both were financially sponsored by Butantan Foundation.

### Subjects

For these two active surveillance studies, children between six months to five years old, pregnant women, post-partum women (mothers up to 45 days after delivery), healthcare professionals, and elderly (persons > 60 years old), which are target groups for annual influenza vaccination defined by the Brazilian National Immunization Program, were invited to participate at the time they presented spontaneously at the study sites. Participants were screened based on the following inclusion and exclusion criteria: inclusion criteria-belong to one of the target groups described above, have an indication to be vaccinated against influenza; be available to participate in the study for its entire duration (6 weeks after administration of any dose of the vaccine), and show intention to participate in the study, documented with the signature of the Informed Consent Form by the subject or a legal representative in case of minors; exclusion criteria—history of allergy to eggs or any other component of the vaccine, suspected or confirmed fever within three days pre-vaccination or axillary temperature over 37.8˚C on the vaccination day; history of severe adverse reaction after previous administration of an Influenza vaccine within 6 weeks following vaccination, history of Guillain-Barré Syndrome or other demyelinating neuropathies, behavioral, cognitive or psychiatric disease that in the opinion of the principal investigator or his representative physician, affects the participant's ability to understand and cooperate with all study protocol requirements, any other condition which, in the opinion of the principal investigator or his/her medical delegate warrants the exclusion of the subject from the study. All participants provided signed informed consent and met all the inclusion criteria and none of the exclusion criteria.

### Primary, secondary and exploratory endpoints

The primary safety endpoint was characterization of AEFI during a period of 42 days after administration of any dose of the 2017 and 2018 trivalent influenza vaccines Southern hemisphere vaccines produced by IB regarding incidence, causality, duration, and seriousness. The secondary endpoints were: characterization of solicited AEFI during a period of 14 days after administration of any dose of the vaccine regarding incidence, causality, duration and seriousness; characterization of unsolicited AEFI during a period of 14 days after administration of any dose of the vaccine regarding incidence, causality, duration and seriousness, and characterization of unsolicited AEFI during a period of 42 days after administration of any dose of the vaccine regarding incidence, causality, duration and seriousness per vaccination group.

The exploratory endpoints were: frequency of AEFI according to WHO causality assessment classification, and frequency of neurological signs and/or symptoms associated with Guillain-Barré Syndrome during a period of 42 days after administration of any dose of the vaccine [8].

## Studies procedures for active surveillance

This was an observational prospective cohort study (active pharmacovigilance study), and therefore, included active surveillance of the AEFI that was performed at the study sites as well via telephone contact. The participants were considered enrolled in the study (Day 0) at the time of vaccination. Thirty minutes after vaccination, the study site staff evaluated each participant for the immediate occurrence of solicited and unsolicited AEFIs. Before leaving, participants were instructed to return to the study site for a visit on Day 3 considering 2 extra days tolerance (+2 days), answer a telephone call on Day 7 (+3 days) post-vaccination and return for the final visit on Day 14 (+4 days) post-vaccination, they received the Participant's Diary to be filled with both solicited and unsolicited AEFI occurred within the 14 days post- vaccination, and were instructed to inform the study staff immediately whenever there was hospitalization and/or serious adverse events. The participants were also instructed on how to record the potential AEFI and to perform severity/intensity classifications of the AEFI using a simple scoring system. A final telephone contact was made six weeks after vaccination, to register the occurrence of other potential adverse events, such as Guillain-Barré Syndrome (GBS). For children who received two doses of the vaccine solicited and unsolicited AEFI data were collected after the first and second doses following the same procedures. According to the Brazilian pharmacovigilance legislation [9], all pregnant women enrolled in this study were followed according to the IB Pharmacovigilance routine procedures, which included contacts via telephone calls in every trimester of the pregnancy after vaccination and a final contact after the probable date of birth to check the baby's health status. If the child was born healthy then the follow-up was terminated, otherwise, the child was followed up to assess adverse event causality. In the event of a miscarriage, a retrospective investigation of the case was performed by using the available medical data.

## Safety assessment

The definitions of adverse events followed the recommendations of the Good Clinical Practices [7], the Guideline on Clinical Safety Data Management from the International Conference on Harmonization (ICH) [10] the Brighton Collaboration case definitions [11], and when case definitions were not available on the Brighton Collaboration sources, the Common Terminology Criteria for Adverse Events (CTCAE) of the US Department of Health and Human Services [12] were used. In summary, an adverse event was defined as any untoward medical occurrence affecting a vaccinated participant and that does not necessarily have a causal relationship to the vaccine. Therefore, an adverse event could be any unfavorable and unintended sign, symptom, or disease (including an abnormal laboratory finding), temporally associated with the vaccination. All adverse events were encoded and grouped according to the MedDRA methodology Medical Dictionary for Regulatory Activities) [13]. A Serious Adverse Event (SAE) was defined as any adverse event that results in any of the following outcomes: death, life threatening condition (one that, according to the evaluation of the notifier puts the individual in immediate danger of death due to the adverse event); requirement of hospitalization or prolonging a current hospitalization, significant or persistent incapacity (substantial interruption of a person's ability to lead the normal life functions); congenital anomaly, any suspected transmission of infectious agent through a drug, and a clinically significant adverse event (adverse event arising from the use of medications requiring medical intervention in

order to avoid death, life threatening, significant disability or hospitalization). An Adverse Reaction (AR) was defined as any AE with a reasonably causal relationship with the vaccine, as determined by the adapted classification of "Uppsala Monitoring Centre" (UMC) of the World Health Organization (WHO) [14–16]. The intensity of AE was classified as grade one to four according to the Toxicity Grading Scale for Healthy Adult and Adolescent Volunteers Enrolled in Preventive Vaccine Clinical Trials of the US Food and Drug Administration [17]. The duration of AE was recorded in days. All local adverse events were considered as adverse reactions to IB influenza vaccine. The study staffs were instructed to include in their adverse events evaluation the following information: classification of seriousness; classification of intensity [18] (Annex AI), causality assessment [8] (Annex AII), and adverse event outcome.

## Study vaccines

The composition of seasonal the 2017 and 2018 TIVs evaluated in the studies followed the WHO composition recommendation for the Southern Hemisphere. The compositions in the 2017 campaign (Centro Buritis: Lot IB 170047 and Centro CRIE Unifesp: Lot IB 170074): influenza A/Michigan/45/2015 (H1N1) pdm09-like virus, influenza A/Hong Kong/4801/2014 (H3N2)-like virus, and influenza B/Brisbane/60/2008-like virus; and 2018 campaign (Centro CRIE Unifesp: Lots IB 18069 and IB 180076): influenza A/Michigan/45/2015 (H1N1) pdm09-like virus; influenza A/Singapore/INFIMH-16-0019/2016 (H3N2)-like virus, and influenza B/Phuket/3073/2013-like virus. The vaccines were formulated in a 10-dose vials (0.5mL per dose) and stored at 2–8˚C.

## Sample size

According to EMA guidelines [7] on strengthening surveillance of seasonal influenza vaccine safety in the European Union, at least 100 individuals from each of the target groups for vaccination should be monitored for AEFI. For the 2017 active pharmacovigilance study, a total sample of 900 participants was planned to be evaluated for solicited and unsolicited AEFI through active surveillance. Participants were divided in the following study targeted groups: 100 children between six months and five years old, 100 pregnant women and 100 postpartum women, 300 health care workers and 300 elderly. The sample size of 900 would allow the study to detect adverse events with frequency equal to or higher than 0.6%, while a sample size of 300 would allow to detect an event with frequency equal to or greater than 1%. The 2017 study was performed over the 2017 Brazilian Influenza National Immunization Campaign, i.e. in limited vaccination period window. Hence, the total planned sample sizes for some of the target groups were not reached: elderly (n = 55/300), health professionals (n = 235/300), and postpartum women (n = 33/100). On the other hand, the number of participants in the children and pregnant women groups was slightly higher than the number planed: children (109/100) and pregnant women (108/100). To increase safety data regarding seasonal influenza vaccination in elderly and health professionals, we conducted the 2018 active pharmacovigilance study with a total number of participants of 400: 250 elderly and 150 health professionals. Postpartum women were not considered for the 2018 study because of the lower vaccine coverage observed for that target group in the previous year.

## Statistical analysis

A descriptive statistical data analysis of AR was performed, stratified by study groups (elderly, children, healthcare workers, pregnant and postpartum women) [19]. Results are presented as frequency of occurrence (absolute and percentage values) for qualitative variables and as dispersion measures (quartiles) for quantitative variables. The frequency of solicited and

unsolicited ARs were calculated along with percentile limits. The software Stata 13.0 (Stata-Corp LP, College Station, Texas USA) was used for the statistical analyses.

## 3. Results

### Participants

The total number of invited participants from the 2017 and 2018 studies was 949 (546 and 403 from the 2017 and 2018 studies, respectively) (Fig 1). Seven (7) out of 949 invited participants were excluded from the study (4 unavailable for the entire study period, 1 no indication for vaccination, and 2 unknown reasons). Among the 942 enrolled participants, 305 were elderly, 109 children, 108 pregnant women, 32 postpartum women, and 388 healthcare workers.

The demographic characteristics of the participants are described in Table 1. The median age of the participants was: elderly 68.0 years, children 2.1 years, pregnant women 26.2 years, postpartum women 27.5 years, and healthcare workers 24.9 years. Most participants were female 69.5%, and white 55.5%.

### Adverse events and adverse reactions

Considering the primary endpoint, the distribution of adverse events (AEs) 42 days postvaccination classified as adverse reactions (ARs) according to WHO are described in Table 2. The total number of AEs were 2,147, of those 1,643 were AR. The overall median of AEs per participant was 2, and among elderly, children, pregnant women, postpartum women, and health workers was 2, 3, 4, 4, 2, and 2, respectively. The overall median of ARs per participant was 2, and among elderly, children, pregnant women, postpartum women, and health workers was 1,

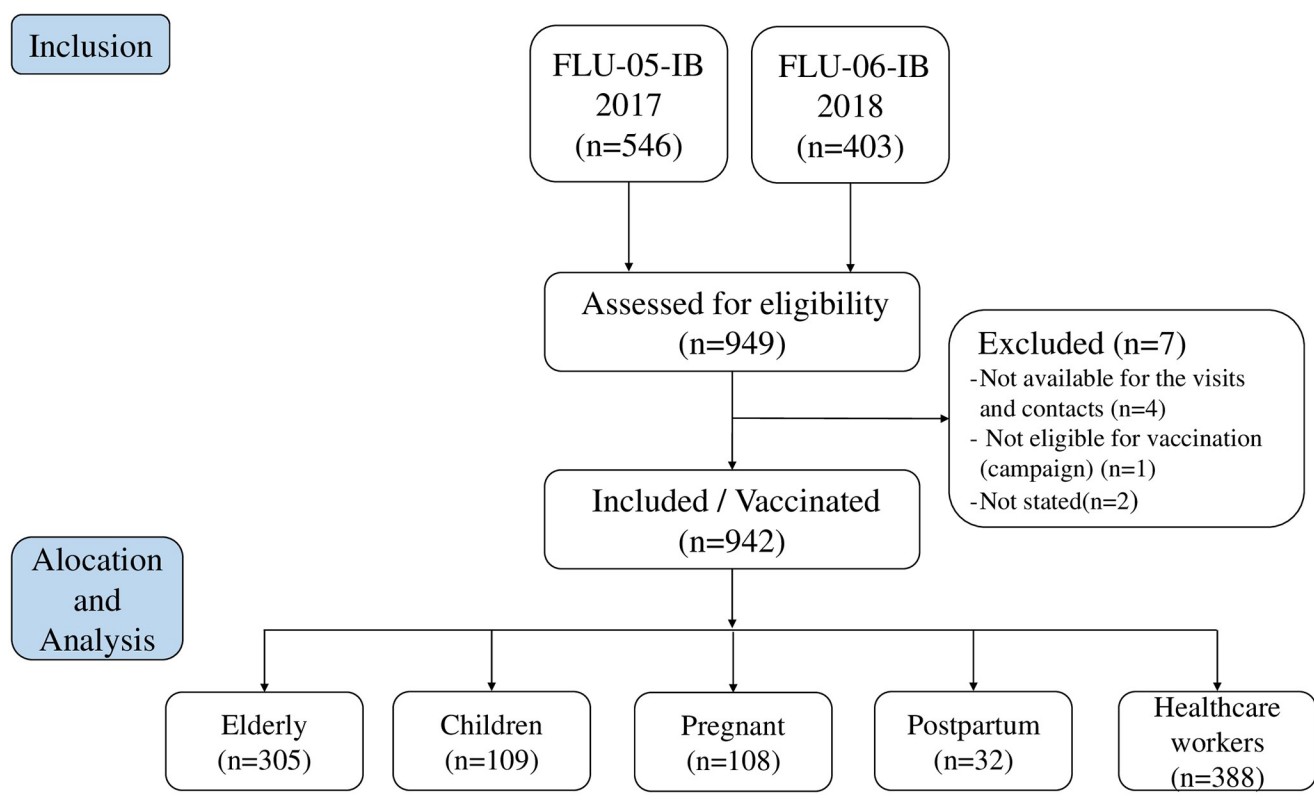

**Fig 1. Study participants algorithm: Screening, enrollment, and vaccination.**

**Table 1. Demographic characteristics of all participants by study group in both studies (2017 and 2018).**

| VARIABLES | Elderly (n = 305) | Children (n = 109) | Pregnant women (n = 108) | Postpartum women (n = 32) | Healthcare workers (n = 388) | TOTAL (n = 942) |
|---|---|---|---|---|---|---|
| **Age** ($P_{50}$ ($P_{25}$– $P_{75}$)) | 68.0 (64–73.6) | 2.1 (1.2–3.4) | 26.2 (22.7–30.8) | 27.5 (20.9–31.8) | 24.9 (22.8–30.3) | 28.1 (22.5–63.6) |
| **Sex [n (%)]** | | | | | | |
| Female | 204 (66.9) | 55 (50.5) | 108 (100) | 32 (100) | 256 (66.0) | 655 (69.5) |
| Male | 101 (33.1) | 54 (49.5) | - | - | 132 (34.0) | 287 (30.5) |
| **Ethnicity [n (%)]** | | | | | | |
| White | 224 (73.4) | 24 (22.0) | 10 (9.3) | 1 (3.1) | 264 (68.0) | 523 (55.5) |
| Black | 13 (4.3) | 3 (2.8) | 4 (3.7) | 2 (6.3) | 19 (4.9) | 41 (4.4) |
| Multiracial | 20 (6.6) | 76 (69.7) | 89 (82.4) | 29 (90.6) | 67 (17.3) | 281 (29.8) |
| Asian | 47 (15.4) | 1 (0.9) | 2 (1.8) | - | 37 (9.5) | 87 (9.2) |
| Other | 1 (0.3) | 5 (4.6) | 3 (2.8) | - | 1 (0.3) | 10 (1.1) |
| **Clinical Sites [n (%)]** | | | | | | |
| CRIE UNIFESP | 287 (94.1) | | 1 (0.9) | - | 358 (92.3) | 646 (68.6) |
| Buritis | 18 (5.9) | 109 (100) | 107 (99.1) | 32 (100) | 30 (7.7) | 296 (31.4) |

$P_{50}$: Median; $P_{25}$: Lower quartile; $P_{75}$: Upper quartile.

2, 4, 2, 1.5, and 2, respectively. As showed in the supplementary material 2, most AEs and ARs were grade 1/2 and lasted less than a day.

Considering the exploratory endpoint, no symptoms associated with GBS were reported during the entire period of the studies 2017 and 2018 in any population investigated.

## Solicited adverse reaction

**Solicited adverse reaction in elderly.** Considering the secondary endpoints, overall 44.3% of the elderly presented with solicited ARs. The most frequent local and systemic adverse reactions were pain (28.5%) and headache (11.1%), respectively. The majority of local and systemic solicited ARs were classified as Grade 1 or Grade 2. Fewer episodes of local

**Table 2. Distribution of adverse events (AE) during the entire study (42 days) and adverse reactions (AR) by study group both in 2017 and 2018.**

| GROUP | AE n° | Participants with AE* n° (%) | AE per individual median ($P_{25}$-$P_{75}$) |
|---|---|---|---|
| Elderly (n = 305) | 445 | 187 (61.3) | 2 (1–3) |
| Children (n = 109) | 327 | 75 (68.8) | 3 (2–5) |
| Pregnant women (n = 108) | 436 | 82 (75.9) | 4 (2–7) |
| Postpartum women (n = 32) | 95 | 22 (68.8) | 4 (2–8) |
| Healthcare workers (n = 388) | 844 | 313 (80.7) | 2 (1–3) |
| **TOTAL (n = 942)** | **2,147** | **679 (72.1)** | **2 (1–4)** |
| **GROUP** | **AR n°** | **Participants with AR n° (%)** | **AR per individual median ($P_{25}$-$P_{75}$)** |
| Elderly (n = 305) | 311 | 141 (46.2) | 1 (1–3) |
| Children (n = 109) | 252 | 72 (66.1) | 2 (1–5) |
| Pregnant women (n = 108) | 375 | 78 (72.2) | 4 (2–7) |
| Postpartum women (n = 32) | 73 | 22 (68.8) | 2 (1–6) |
| Healthcare workers (n = 388) | 642 | 298 (76.8) | 1.5 (1–2) |
| **TOTAL (n = 942)** | **1,653** | **611 (64.9)** | **2 (1–3)** |

Adverse Reaction (AR) was defined as any AE with a reasonably causal relationship with the vaccine, according to UMC-WHO.

**Table 3. Frequency and intensity of solicited adverse reactions (AR) in elderly, observed both in 2017 and 2018.**

| | Elderly with AR (n = 305) | | ARs Total | Grade 1 | | Grade 2 | | Grade 3 | | Grade 4 | |
|---|---|---|---|---|---|---|---|---|---|---|---|
| | n | % | n | n | % | n | % | n | % | n | % |
| **SOLICITED** | | | | | | | | | | | |
| **Local** | | | | | | | | | | | |
| Pain | 87 | 28.5 | 89 | 86 | 96.6 | 3 | 3.4 | 0 | 0.0 | 0 | 0.0 |
| Ecchymosis | 7 | 2.3 | 7 | 6 | 85.7 | 1 | 14.3 | 0 | 0.0 | 0 | 0.0 |
| Erythema | 9 | 3.0 | 9 | 6 | 66.7 | 2 | 22.2 | 1 | 11.1 | 0 | 0.0 |
| Swelling | 13 | 4.3 | 15 | 7 | 46.7 | 5 | 33.3 | 3 | 20.0 | 0 | 0.0 |
| Induration | 10 | 3.3 | 10 | 7 | 70.0 | 3 | 30.0 | 0 | 0.0 | 0 | 0.0 |
| Pruritus | 9 | 3.0 | 9 | 8 | 88.9 | 1 | 11.1 | 0 | 0.0 | 0 | 0.0 |
| **Systemic** | | | | | | | | | | | |
| Arthralgia | 12 | 3.9 | 12 | 11 | 91.7 | 1 | 8.3 | 0 | 0.0 | 0 | 0.0 |
| Chills | 9 | 3.0 | 9 | 9 | 100 | 0 | 0.0 | 0 | 0.0 | 0 | 0.0 |
| Headache | 34 | 11.1 | 38 | 26 | 68.4 | 11 | 28.9 | 1 | 2.6 | 0 | 0.0 |
| Persistent crying | 0 | 0.0 | 0 | 0 | 0.0 | 0 | 0.0 | 0 | 0.0 | 0 | 0.0 |
| Rash | 0 | 0.0 | 0 | 0 | 0,0 | 0 | 0.0 | 0 | 0.0 | 0 | 0.0 |
| Fatigue | 29 | 9.5 | 34 | 30 | 88.2 | 3 | 8.8 | 1 | 2.9 | 0 | 0.0 |
| Fever | 1 | 0.3 | 1 | 1 | 100 | 0 | 0.0 | 0 | 0.0 | 0 | 0.0 |
| Irritability | 0 | 0.0 | 0 | 0 | 0.0 | 0 | 0.0 | 0 | 0.0 | 0 | 0.0 |
| Malaise | 14 | 4.6 | 16 | 12 | 75.0 | 3 | 18.8 | 1 | 6.3 | 0 | 0.0 |
| Myalgia | 26 | 8.5 | 26 | 21 | 80.8 | 5 | 19.2 | 0 | 0.0 | 0 | 0.0 |
| Pruritus | 0 | 0.0 | 0 | 0 | 0.0 | 0 | 0.0 | 0 | 0.0 | 0 | 0.0 |
| **TOTAL** | 135 | 44.3 | 275 | 230 | 83.6 | 38 | 13.8 | 7 | 2.5 | 0 | 0.0 |

Adverse Reaction (AR) was defined as any AE with a reasonably causal relationship with the vaccine, according to UMC-WHO. Solicited ARs were considered up to 14 days postvaccination.

erythema (1), local swelling (3), headache (1), fatigue (1), and malaise (1) were classified as Grade 3. No local and systemic solicited ARs were classified as Grade 4 (Table 3).

**Solicited adverse reaction in children.** Considering the secondary endpoints, overall 60.6% of the children presented with solicited ARs. The most frequent local AR was pain (22.9%) and systemic ARs were fever (26.6%) and persistent crying (21.1%). The majority of local and systemic solicited ARs were classified as Grade 1 or Grade 2. Fewer episodes of headache (3), persistent crying (3), fever (1), and malaise (2) were classified as Grade 3. Samewise, fewer episodes of headache (1), persistent crying (1), fatigue (1), irritability (2), and malaise (1) were classified as Grade 4 (Table 4).

**Solicited adverse reaction in pregnant women.** Considering the secondary endpoints, overall 71.3% of the pregnant women presented with solicited ARs. The most frequent local AR was pain (49.1%) and systemic ARs were headache (46.3%), malaise (40.7%), and fatigue (36.1%). The majority of local and systemic solicited ARs were classified as Grade 1 or Grade 2. Fewer episodes of local pain (2), chills (1), headache (6), fatigue (4), malaise (4), and myalgia (2) were classified as Grade 3. Additionally, fewer episodes of fatigue (1) and malaise (1) were classified as Grade 4 (Table 5).

**Solicited adverse reaction in postpartum women.** Considering the secondary endpoints, overall 65.6% of the postpartum women presented with solicited ARs. The most frequent local AR was pain (31.3%) and systemic ARs were headache (43.8%), fever (25.0%), and fatigue (25.0%). The majority of local and systemic solicited ARs were classified as Grade 1 or Grade

**Table 4. Frequency and intensity of solicited adverse reactions (AR) in children, observed in both 2017 and 2018.**

| | Children with AR (n = 109) | | ARs Total | Grade 1 | | Grade 2 | | Grade 3 | | Grade 4 | |
|---|---|---|---|---|---|---|---|---|---|---|---|
| | n | % | n | n | % | n | % | n | % | n | % |
| **SOLICITED** | | | | | | | | | | | |
| **Local** | | | | | | | | | | | |
| Pain | 25 | 22.9 | 25 | 22 | 88.0 | 3 | 12.0 | 0 | 0.0 | 0 | 0.0 |
| Ecchymosis | 3 | 2.8 | 3 | 3 | 100 | 0 | 0.0 | 0 | 0.0 | 0 | 0.0 |
| Erythema | 4 | 3.7 | 4 | 4 | 100 | 0 | 0.0 | 0 | 0.0 | 0 | 0.0 |
| Swelling | 2 | 1.8 | 2 | 2 | 100 | 0 | 0.0 | 0 | 0.0 | 0 | 0.0 |
| Induration | 2 | 1.8 | 2 | 2 | 100 | 0 | 0.0 | 0 | 0.0 | 0 | 0.0 |
| Pruritus | 3 | 2.8 | 3 | 3 | 100 | 0 | 0.0 | 0 | 0.0 | 0 | 0.0 |
| **Systemic** | | | | | | | | | | | |
| Arthralgia | 4 | 3.7 | 4 | 4 | 100 | 0 | 0.0 | 0 | 0.0 | 0 | 0.0 |
| Chills | 3 | 2.8 | 3 | 2 | 66.7 | 1 | 33.3 | 0 | 0.0 | 0 | 0.0 |
| Headache | 20 | 18.3 | 26 | 17 | 65.4 | 5 | 19.2 | 3 | 11.5 | 1 | 3.8 |
| Persistent crying | 23 | 21.1 | 35 | 20 | 57.1 | 11 | 31.4 | 3 | 8.6 | 1 | 2.9 |
| Rash | 4 | 3.7 | 4 | 4 | 100 | 0 | 0.0 | 0 | 0.0 | 0 | 0.0 |
| Fatigue | 12 | 11.0 | 16 | 14 | 87.5 | 1 | 6.3 | 0 | 0.0 | 1 | 6.3 |
| Fever | 29 | 26.6 | 31 | 24 | 77.4 | 6 | 19.4 | 1 | 3.2 | 0 | 0.0 |
| Irritability | 13 | 11.9 | 19 | 11 | 57.9 | 6 | 31.6 | 0 | 0.0 | 2 | 10.5 |
| Malaise | 20 | 18.3 | 23 | 16 | 69.6 | 4 | 17.4 | 2 | 8.7 | 1 | 4.3 |
| Myalgia | 11 | 10.1 | 12 | 7 | 58.3 | 5 | 41.7 | 0 | 0.0 | 0 | 0.0 |
| Pruritus | 1 | 0.9 | 1 | 1 | 100 | 0 | 0.0 | 0 | 0.0 | 0 | 0.0 |
| **TOTAL** | 66 | 60.6 | 213 | 156 | 73.2 | 42 | 19.7 | 9 | 4.2 | 6 | 2.8 |

Adverse Reaction (AR) was defined as any AE with a reasonably causal relationship with the vaccine, according to UMC-WHO. Solicited ARs were considered up to 14 days postvaccination.

2. Fewer episodes of chills (2) and headache (2) were classified as Grade 3. No local and systemic solicited ARs were classified as Grade 4 (Table 6).

**Solicited adverse reaction in healthcare workers.** Considering the secondary endpoints, overall 76.0% of the healthcare workers presented with solicited ARs. The most frequent local AR was pain (68.0%) and systemic ARs were headache (24.2%) and fatigue (16.2%). The majority of local and systemic solicited ARs were classified as Grade 1 or Grade 2. Fewer episodes of headache (2) and myalgia (1) were classified as Grade 3. Likewise, fewer episodes of local erythema (1), local swelling (1), headache (1), and malaise (1) were classified as Grade 4 (Table 7).

## Distribution and characteristics of solicited Adverse Reactions (ARs) in the entire study population

Considering the secondary endpoints, overall 594 (63.1%) participants presented with at least one adverse reaction (AR) and medication was required for 12.9% of the AR. The local AR median onset time was 1 day and for the systemic AR varied from 1 to 5 days. The local and systemic AR median duration was 1 day (Table 8).

## Unsolicited adverse reactions

Considering the secondary endpoints, the overall frequency and intensity of the unsolicited ARs can be found at the supplement for elderly (Table S-1 in S1 File), children (Table S-2 in

**Table 5.** Frequency and intensity of solicited adverse reactions (AR) in pregnant women, observed in both 2017 and 2018.

| | Pregnant with AR (n = 108) | | ARs Total | Grade 1 | | Grade 2 | | Grade 3 | | Grade 4 | |
|---|---|---|---|---|---|---|---|---|---|---|---|
| | n | % | n | n | % | n | % | n | % | n | % |
| **SOLICITED** | | | | | | | | | | | |
| **Local** | | | | | | | | | | | |
| Pain | 53 | 49.1 | 56 | 47 | 83.9 | 7 | 12.5 | 2 | 3.6 | 0 | 0.0 |
| Ecchymosis | 2 | 1.9 | 2 | 1 | 50.0 | 1 | 50.0 | 0 | 0.0 | 0 | 0.0 |
| Erythema | 3 | 2.8 | 3 | 3 | 100 | 0 | 0.0 | 0 | 0.0 | 0 | 0.0 |
| Swelling | 5 | 4.6 | 5 | 4 | 80.0 | 1 | 20.0 | 0 | 0.0 | 0 | 0.0 |
| Induration | 4 | 3.7 | 4 | 4 | 100 | 0 | 0.0 | 0 | 0.0 | 0 | 0.0 |
| Pruritus | 7 | 6.5 | 9 | 8 | 88.9 | 1 | 11.1 | 0 | 0.0 | 0 | 0.0 |
| **Systemic** | | | | | | | | | | | |
| Arthralgia | 19 | 17.6 | 24 | 15 | 62.5 | 9 | 37.5 | 0 | 0.0 | 0 | 0.0 |
| Chills | 9 | 8.3 | 10 | 5 | 50.0 | 4 | 40.0 | 1 | 10.0 | 0 | 0.0 |
| Headache | 50 | 46.3 | 84 | 57 | 67.9 | 21 | 25.0 | 6 | 7.1 | 0 | 0.0 |
| Persistent crying | 0 | 0.0 | 0 | 0 | 0.0 | 0 | 0.0 | 0 | 0.0 | 0 | 0.0 |
| Rash | 2 | 1.9 | 2 | 2 | 100 | 0 | 0.0 | 0 | 0.0 | 0 | 0.0 |
| Fatigue | 39 | 36.1 | 53 | 36 | 67.9 | 12 | 22.6 | 4 | 7.5 | 1 | 1.9 |
| Fever | 5 | 4.6 | 6 | 4 | 66.7 | 2 | 33.3 | 0 | 0.0 | 0 | 0.0 |
| Irritability | 0 | 0.0 | 0 | 0 | 0.0 | 0 | 0.0 | 0 | 0.0 | 0 | 0.0 |
| Malaise | 44 | 40.7 | 53 | 36 | 67.9 | 12 | 22.6 | 4 | 7.5 | 1 | 1.9 |
| Myalgia | 17 | 15.7 | 21 | 16 | 76.2 | 3 | 14.3 | 2 | 9.5 | 0 | 0.0 |
| Pruritus | 0 | 0.0 | 0 | 0 | 0.0 | 0 | 0.0 | 0 | 0.0 | 0 | 0.0 |
| **TOTAL** | 77 | 71.3 | 332 | 238 | 71.7 | 73 | 22.0 | 19 | 5.7 | 2 | 0.6 |

Adverse Reaction (AR) was defined as any AE with a reasonably causal relationship with the vaccine, according to UMC-WHO. Solicited ARs were considered up to 14 days postvaccination.

S1 File), pregnant women (Table S-3 in S1 File), postpartum women (Table S-4 in S1 File), healthcare workers (Table S-5 in S1 File) and entire study population (Table S-6 in S1 File). The most frequent reported unsolicited ARs were musculoskeletal pain, generalized itching, and cough. It was reported only 1 Grade 4 unsolicited AR (musculoskeletal pain) in a child, 1 Grade 3 (somnolence) in healthcare worker, and 2 Grade 3 (musculoskeletal pain) in pregnant women. All unsolicited ARs in elderly and in postpartum women were classified as either Grade 1 or 2 (in postpartum women they were all classified as Grade 2). No serious adverse events were reported in the study.

## 4. Discussion

The Ministry of Health (MoH) of Brazil, through its National Immunization Program (NIP), has promoted annual national influenza vaccination campaigns since 1999. In 2013, IB released the first batches of seasonal influenza vaccines entirely produced by the Institute. From 2013 to 2019 approximately 260 million doses of TIV were produced and delivered to the Brazilian MoH. Passive pharmacovigilance activities for seasonal influenza vaccines have been performed by IB since 2013, and therefore, the results of the studies presented herein reflect the first experience of IB in performing active pharmacovigilance activities for its seasonal influenza vaccine.

A total of 949 volunteers participated in the 2017 and 2018 studies (546 in 2017 and 403 in 2018). Among them, 305 were elderly, 109 children, 108 pregnant women, 32 postpartum

**Table 6. Frequency and intensity of solicited adverse reactions (AR) in postpartum women, observed in both 2017 and 2018.**

| | Postpartum with AR (n = 32) | | ARs Total | Grade 1 | | Grade 2 | | Grade 3 | | Grade 4 | |
|---|---|---|---|---|---|---|---|---|---|---|---|
| | n | % | n | n | % | n | % | N | % | n | % |
| **SOLICITED** | | | | | | | | | | | |
| **Local** | | | | | | | | | | | |
| Pain | 10 | 31.3 | 10 | 9 | 90.0 | 1 | 10.0 | 0 | 0.0 | 0 | 0.0 |
| Ecchymosis | 0 | 0.0 | 0 | 0 | 0.0 | 0 | 0.0 | 0 | 0.0 | 0 | 0.0 |
| Erythema | 0 | 0.0 | 0 | 0 | 0.0 | 0 | 0.0 | 0 | 0.0 | 0 | 0.0 |
| Swelling | 0 | 0.0 | 0 | 0 | 0.0 | 0 | 0.0 | 0 | 0.0 | 0 | 0.0 |
| Induration | 2 | 6.3 | 2 | 2 | 100 | 0 | 0.0 | 0 | 0.0 | 0 | 0.0 |
| Pruritus | 0 | 0.0 | 0 | 0 | 0.0 | 0 | 0.0 | 0 | 0.0 | 0 | 0.0 |
| **Systemic** | | | | | | | | | | | |
| Arthralgia | 2 | 6.3 | 2 | 1 | 50.0 | 1 | 50.0 | 0 | 0.0 | 0 | 0.0 |
| Chills | 4 | 12.5 | 5 | 3 | 60.0 | 0 | 0.0 | 2 | 40.0 | 0 | 0.0 |
| Headache | 14 | 43.8 | 25 | 20 | 80.0 | 3 | 12.0 | 2 | 8.0 | 0 | 0.0 |
| Persistent crying | 0 | 0.0 | 0 | 0 | 0.0 | 0 | 0.0 | 0 | 0.0 | 0 | 0.0 |
| Rash | 0 | 0.0 | 0 | 0 | 0.0 | 0 | 0.0 | 0 | 0.0 | 0 | 0.0 |
| Fatigue | 8 | 25.0 | 10 | 10 | 100 | 0 | 0.0 | 0 | 0.0 | 0 | 0.0 |
| Fever | 8 | 25.0 | 9 | 9 | 100 | 0 | 0.0 | 0 | 0.0 | 0 | 0.0 |
| Irritability | 0 | 0.0 | 0 | 0 | 0.0 | 0 | 0.0 | 0 | 0.0 | 0 | 0.0 |
| Malaise | 3 | 9.4 | 4 | 1 | 25.0 | 3 | 75.0 | 0 | 0.0 | 0 | 0.0 |
| Myalgia | 2 | 6.3 | 2 | 1 | 50.0 | 1 | 50.0 | 0 | 0.0 | 0 | 0.0 |
| Pruritus | 0 | 0.0 | 0 | 0 | 0.0 | 0 | 0.0 | 0 | 0.0 | 0 | 0.0 |
| **TOTAL** | 21 | 65.6 | 69 | 56 | 81.2 | 9 | 13.0 | 4 | 5.8 | 0 | 0.0 |

Adverse Reaction (AR) was defined as any AE with a reasonably causal relationship with the vaccine, according to UMC-WHO. Solicited ARs were considered up to 14 days postvaccination.

women, and 388 healthcare workers. The total number of AEs and ARs were 2,147 and 1,643, respectively. The overall median of AEs and of ARs per participant was 2.

Overall, 44.3% of the elderly, 60.6% of the children, 71.3% of the pregnant women, 65.6% of the postpartum women, and 76.0% of the healthcare workers presented with solicited ARs. The comparison of AR frequency among subgroups is limited by different characteristics and sample sizes. The most frequent local AR was local pain, headache, malaise, and fatigue. In the enhanced safety surveillance of influenza vaccines in Europe from 2014 to 2017, which assessed safety of several influenza vaccine brands from Seqirus, Sanofi Pasteur, GSK and AstraZeneca in different age groups, including one high risk conditions group [20], local pain and headache were also cited as the most frequently reported ARs. At the 42 days post immunization, no neurological signs and/or symptoms associated with GBS were reported in any of the five study groups.

Fever was most frequent AR among children (26.6%) and postpartum women (25.5%). In the Pillsbury et al [21] enhanced safety surveillance of trivalent influenza vaccines—Vaxigrip, Fluarix, Influvac and Agrippal (Sanofi-Pasteur, GlaxoSmithKline, BGP Products and Novartis manufacturers, respectively) in Australia, fever was observed in 4.4% of children. However, the authors reported solicited adverse events up to 3 days post immunization only, whereas our study followed up the participants for solicited adverse events post immunization up to 14 days. In children, persistent crying (21.1%) was the second most frequent solicited AR.

**Table 7. Frequency and intensity of solicited adverse reactions (AR) in healthcare workers, observed in both 2017 and 2018.**

| | HC work with AR (n = 388) | | ARs Total | Grade 1 | | Grade 2 | | Grade 3 | | Grade 4 | |
|---|---|---|---|---|---|---|---|---|---|---|---|
| | n | % | n | n | % | n | % | n | % | n | % |
| **SOLICITED** | | | | | | | | | | | |
| **Local** | | | | | | | | | | | |
| Pain | 264 | 68.0 | 270 | 251 | 93.0 | 19 | 7.0 | 0 | 0.0 | 0 | 0.0 |
| Ecchymosis | 2 | 0.5 | 2 | 2 | 100 | 0 | 0.0 | 0 | 0.0 | 0 | 0.0 |
| Erythema | 7 | 1.8 | 7 | 6 | 85.7 | 0 | 0.0 | 0 | 0.0 | 1 | 14.3 |
| Swelling | 9 | 2.3 | 9 | 8 | 88.9 | 0 | 0.0 | 0 | 0.0 | 1 | 11.1 |
| Induration | 14 | 3.6 | 14 | 12 | 85.7 | 2 | 14.3 | 0 | 0.0 | 0 | 0.0 |
| Pruritus | 16 | 4.1 | 16 | 16 | 100 | 0 | 0.0 | 0 | 0.0 | 0 | 0.0 |
| **Systemic** | | | | | | | | | | | |
| Arthralgia | 17 | 4.4 | 18 | 15 | 83.3 | 3 | 16.7 | 0 | 0.0 | 0 | 0.0 |
| Chills | 12 | 3.1 | 12 | 11 | 91.7 | 1 | 8.3 | 0 | 0.0 | 0 | 0.0 |
| Headache | 94 | 24.2 | 114 | 78 | 68.4 | 33 | 28.9 | 2 | 1.8 | 1 | 0.9 |
| Persistent crying | 0 | 0.0 | 0 | 0 | 0.0 | 0 | 0.0 | 0 | 0.0 | 0 | 0.0 |
| Rash | 1 | 0.3 | 1 | 0 | 0.0 | 1 | 100 | 0 | 0.0 | 0 | 0.0 |
| Fatigue | 63 | 16.2 | 73 | 58 | 79.5 | 15 | 20.5 | 0 | 0.0 | 0 | 0.0 |
| Fever | 2 | 0.5 | 2 | 1 | 50.0 | 1 | 50.0 | 0 | 0.0 | 0 | 0.0 |
| Irritability | 0 | 0.0 | 0 | 0 | 0.0 | 0 | 0.0 | 0 | 0.0 | 0 | 0.0 |
| Malaise | 23 | 5.9 | 28 | 17 | 60.7 | 10 | 35.7 | 0 | 0.0 | 1 | 3.6 |
| Myalgia | 31 | 8.0 | 35 | 25 | 71.4 | 9 | 25.7 | 1 | 2.9 | 0 | 0.0 |
| Pruritus | 2 | 0.5 | 2 | 1 | 50.0 | 1 | 50.0 | 0 | 0.0 | 0 | 0.0 |
| **TOTAL** | 295 | 76.0 | 603 | 501 | 83.1 | 95 | 15.8 | 3 | 0.5 | 4 | 0.7 |

Solicited ARs were considered up to 14 days postvaccination. Adverse Reaction (AR) was defined as any AE with a reasonably causal relationship with the vaccine, according to UMC-WHO.

Plennevaux et al [22] also registered persistent crying in children (6–23 months) who had received H1N1 vaccine in US. Similar findings were pointed out by Zafazk et al [23] but for those who received DTaP/WCL and DTwP-Hib vacines.

In pregnant and postpartum women who received Butantan vaccine, most of the local and systemic AR were mild and self-limited. Pain at the injection site (49.1%) and headache (46.3%) predominated as local and systemic ARs respectively. These results are compatible with a systematic review study on safety and immunogenicity of TIV in pregnant women published by Munoz et al. [24], but differ slightly from those of Lin et al [25] who presented higher rates of local pain (69.6%) and myalgia (26.1%) as the most frequent ARs in pregnant women. The results of our study may contribute to the reporting and pooling of data vaccine safety during pregnancy on continental and global level [26].

Interestingly, musculoskeletal pain was reported as an unsolicited AR in 33% of the participants. Initially addressed as a safety signal, it was later found that error in medical coding leaded to this finding (myalgia, an expected AR, was miscoded as musculoskeletal pain, an unsolicited and unexpected AR).

Additionally, the majority of local and systemic solicited ARs were classified as Grade 1 or Grade 2 in all study groups. Few episodes of Grade 4 ARs were reported in children, pregnant women, and healthcare workers. Furthermore, 63.1% of all participants presented with at least one AR and medication was required for 12.9% of the AR. The local AR median onset time was

**Table 8. Distribution and characteristics of solicited Adverse Reactions (ARs) in the entire study population, in both 2017 and 2018.**

| Solicited Adverse Reactions | Participants[#] with ARs[*] | | Total ARs[*] | AR needing medication | | ARs[*] duration (days) | | ARs[*] onset time (days) | |
|---|---|---|---|---|---|---|---|---|---|
| | n | % | n | n | % | Median P$_{50}$ | P$_{25}$—P$_{75}$[&] | Median P$_{50}$ | P$_{25}$- P$_{75}$[&] |
| **Local** | | | | | | | | | |
| Pain | 439 | 46.6 | 450 | 11 | 2.4 | 1 | 0–1 | 1 | 0–1 |
| Ecchymosis | 14 | 1.5 | 14 | 0 | 0.0 | 1 | 0–4 | 1 | 1–2 |
| Erythema | 23 | 2.4 | 23 | 0 | 0.0 | 0 | 0–1 | 1 | 0–1 |
| Swelling | 29 | 3.1 | 31 | 0 | 0.0 | 0 | 0–1 | 1 | 1–2 |
| Induration | 32 | 3.4 | 32 | 0 | 0.0 | 0 | 0–1.5 | 1 | 1–2 |
| Pruritus | 35 | 3.7 | 37 | 1 | 2.7 | 0 | 0–1 | 1 | 0–1 |
| **Systemic** | | | | | | | | | |
| Arthralgia | 54 | 5.7 | 60 | 1 | 1.7 | 0 | 0–1 | 2 | 1–6 |
| Chills | 37 | 3.9 | 39 | 2 | 5.1 | 0 | 0–0 | 2 | 1–5 |
| Headache | 212 | 22.5 | 287 | 113 | 39.4 | 0 | 0–1 | 3 | 1–7 |
| Persistent crying | 23 | 2.4 | 35 | 2 | 5.7 | 0 | 0–0 | 2 | 1–6 |
| Rash | 7 | 0.7 | 7 | 0 | 0.0 | 1 | 0–5 | 2 | 1–10 |
| Fatigue | 151 | 16.0 | 186 | 6 | 3.2 | 0 | 0–1 | 2 | 1–5 |
| Fever | 45 | 4.8 | 49 | 41 | 83.7 | 0 | 0–1 | 5 | 1–9 |
| Irritability | 13 | 1.4 | 19 | 0 | 0.0 | 0 | 0–1 | 3 | 1–7 |
| Malaise | 104 | 11.0 | 124 | 4 | 3.2 | 0 | 0–1 | 2 | 1–7 |
| Myalgia | 87 | 9.2 | 96 | 11 | 11.5 | 0 | 0–1 | 2 | 1–6 |
| Pruritus | 3 | 0.3 | 3 | 1 | 33.3 | 0 | 0–1 | 1 | 1–2 |
| **TOTAL SOLICITED** | 594[*] | 63.1 | 1492 | 193 | 12.9 | 0 | 0–1 | 1 | 1–4 |

Adverse Reaction (AR) was defined as any AE with a reasonably causal relationship with the vaccine, according to UMC-WHO. Solicited ARs were considered up to 14 days postvaccination.

[#]Participants who presented with at least one adverse reaction.

[&]P25: Lower quartile; P75: Upper quartile.

1 day and for the systemic AR varied from 1 to 5 days. The local and systemic AR median duration was 1 day.

For decades, post-marketing vaccine safety surveillance has depended on analysis of spontaneous (passive) drug adverse event reporting [27]. Systems such as the Adverse Event Reporting System (FAERS) [28] of the Food and Drug Administration (FDA) in the USA and the World Health Organization (WHO) Programme for International Drug Monitoring [29–31] were established to improve post-marketing surveillance for adverse events (AEs). Those system relies on voluntary reporting by healthcare professionals and/or patients and their families. Brazil has also been depended on analysis of spontaneous vaccine adverse event reporting, and as in many countries, it is required that pharmaceutical and manufactories to report AEs to the national drug regulatory authorities [9]. These spontaneous reporting systems, nevertheless, are hampered by incomplete information and/or under-reporting of events which limit the value of the data, diminishing the ability to establish the real prevalence of ARs due to the difficulty to perform the causality assessment between the product and the reported AE. However, there are many other approaches to evaluate drug/vaccines AEs such as active pharmacovigilance initiatives, which has been encourage by many drug regulatory agencies worldwide.

Unlike passive pharmacovigilance, which is based on voluntary spontaneous reporting of AEs, active pharmacovigilance allows to obtain more detailed safety information, to detect

safety signal faster and accurately, and to avoid AEs under-reporting. Hence, more accurate vaccine safety data from specific target populations may be available. Besides, active pharmacovigilance may contribute to improve vaccination coverage rates and positively address vaccine hesitancy as high-quality vaccine safety information can be available to be communicated to society in a timely manner. Furthermore, active pharmacovigilance initiatives may not only contribute to improve healthcare workers awareness of the importance of reporting vaccine AEs, but also, to educate them on real risks and to point out subpopulations for whom certain vaccines are more beneficial or more harmful (risk-benefit evaluation). Lastly but not least, active pharmacovigilance studies support the vaccine safety monitoring under real-world situations such as usage in specific sub-populations and/or in patients with various co-morbidities.

In conclusion, the results of our studies confirmed the acceptable safety profile of the 2017–2018 TIVs produced by Instituto Butantan in elderly, children, pregnant women, postpartum women. And healthcare workers. The safety profiles described in our study are in accordance with those from other studies evaluating seasonal trivalent influenza vaccine produced by Instituto Butantan [32,33], and with from other studies available in the literature which evaluated similar vaccines such as Vaxigrip [34] and Fluarix [35] and Fluad [36]. This study confirmed the capability of Instituto Butantan to perform active pharmacovigilance studies for its seasonal TIV and it will contribute to the preparedness of Instituto Butantan for the WHO pre-qualification of the seasonal TIV.

## Supporting information

**S1 File.**
(DOCX)

**S2 File.**
(XLSX)

## Acknowledgments

We thank the subjects who volunteered for the study. IB staff were involved with study concept and design, conduct, analysis, and interpretation of the data, drafting of the manuscript, and the decision to submit the manuscript for publication. The corresponding author had full access to the study data and had final responsibility for the decision to submit for publication.

## Author Contributions

**Conceptualization:** Tazio Vanni, Beatriz da Costa Thomé, Mayra Martho Moura de Oliveira, Vera Lúcia Gattás, Patrícia Emília Braga, Roberta de Oliveira Piorelli, Olga Menang, Muriel Soquet, Alexander Roberto Precioso.

**Data curation:** Tazio Vanni, Beatriz da Costa Thomé, Mayra Martho Moura de Oliveira, Vera Lúcia Gattás, Marcelo Eiji Koike, Maria Beatriz Bastos Lucchesi, Patrícia Emília Braga, Roberta de Oliveira Piorelli, Juliana Yukari Koidara Viscondi, Gabriella Mondini, Anderson da Silva, Heloísa Maximo Espínola, Joane do Prado Santos, Samanta Hosokawa Dias de Nóvoa Rocha, Lily Yin Weckx, Olga Menang, Muriel Soquet, Alexander Roberto Precioso.

**Formal analysis:** Tazio Vanni, Beatriz da Costa Thomé, Maria da Graça Salomão, Patrícia Emília Braga, Juliana Yukari Koidara Viscondi, Lily Yin Weckx, Alexander Roberto Precioso.

**Funding acquisition:** Tazio Vanni, Vera Lúcia Gattás, Maria da Graça Salomão, Maria Beatriz Bastos Lucchesi, Alexander Roberto Precioso.

**Investigation:** Tazio Vanni, Beatriz da Costa Thomé, Marcelo Eiji Koike, Patrícia Emília Braga, Roberta de Oliveira Piorelli, Juliana Yukari Koidara Viscondi, Gabriella Mondini, Anderson da Silva, Samanta Hosokawa Dias de Nóvoa Rocha, Lily Yin Weckx, Alexander Roberto Precioso.

**Methodology:** Tazio Vanni, Beatriz da Costa Thomé, Vera Lúcia Gattás, Maria Beatriz Bastos Lucchesi, Patrícia Emília Braga, Juliana Yukari Koidara Viscondi, Gabriella Mondini, Samanta Hosokawa Dias de Nóvoa Rocha, Lily Yin Weckx, Olga Menang, Muriel Soquet.

**Project administration:** Tazio Vanni, Beatriz da Costa Thomé, Anderson da Silva, Heloísa Maximo Espínola, Lily Yin Weckx, Alexander Roberto Precioso.

**Resources:** Anderson da Silva, Heloísa Maximo Espínola, Joane do Prado Santos.

**Software:** Joane do Prado Santos.

**Supervision:** Tazio Vanni, Mayra Martho Moura de Oliveira, Marcelo Eiji Koike, Gabriella Mondini, Anderson da Silva, Heloísa Maximo Espínola, Samanta Hosokawa Dias de Nóvoa Rocha, Lily Yin Weckx, Olga Menang, Muriel Soquet.

**Validation:** Tazio Vanni, Beatriz da Costa Thomé, Maria da Graça Salomão, Patrícia Emília Braga, Roberta de Oliveira Piorelli, Juliana Yukari Koidara Viscondi.

**Writing – original draft:** Tazio Vanni, Maria da Graça Salomão, Patrícia Emília Braga, Juliana Yukari Koidara Viscondi, Alexander Roberto Precioso.

**Writing – review & editing:** Tazio Vanni, Beatriz da Costa Thomé, Mayra Martho Moura de Oliveira, Vera Lúcia Gattás, Marcelo Eiji Koike, Maria Beatriz Bastos Lucchesi, Gabriella Mondini, Anderson da Silva, Samanta Hosokawa Dias de Nóvoa Rocha, Lily Yin Weckx, Olga Menang, Muriel Soquet, Alexander Roberto Precioso.

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
