## [Decision Letter · Decision Letter 0]

19 Nov 2020

PONE-D-20-21550

Active pharmacovigilance of the seasonal trivalent influenza vaccine produced by Instituto Butantan: a prospective cohort study of five target groups

PLOS ONE

Dear Dr. Vanni,

Thank you for submitting your manuscript to PLOS ONE. After careful consideration, we feel that it has merit but does not fully meet PLOS ONE’s publication criteria as it currently stands. Therefore, we invite you to submit a revised version of the manuscript that addresses the points raised during the review process. In particular can the query re novelty be thoroughly addressed as numerous papers already in this field.

We look forward to receiving your revised manuscript.

Kind regards,

Ray Borrow, Ph.D., FRCPath

Academic Editor

PLOS ONE

Journal Requirements:

2. Please provide the manufacturer information and lot numbers of the vaccines.

3. Thank you for stating in the text of your manuscript that you obtained a "signature of the Informed Consent Form by subject or legal representative". Please also add this information to your ethics statement in the online submission form. In addition, please clarify whether parents/guardians gave consent for minors.

"TV, MGS, JYKV, RPG, MTRPC, ROP, PEB, AS, HME, JPS, VLG, MEK, MBBL, MMMO, SHDNR, LYW and ARP are employees of Instituto Butantan. BCT and GM are former employees of Instituto Butantan. SHDNR and LYW received financial aid for their roles as principal investigators."

Reviewers' comments:

Reviewer's Responses to Questions

**Comments to the Author**

1. Is the manuscript technically sound, and do the data support the conclusions?

Reviewer #1: Partly

Reviewer #2: Yes

2. Has the statistical analysis been performed appropriately and rigorously? 

Reviewer #1: Yes

Reviewer #2: Yes

3. Have the authors made all data underlying the findings in their manuscript fully available?

Reviewer #1: Yes

Reviewer #2: Yes

4. Is the manuscript presented in an intelligible fashion and written in standard English?

Reviewer #1: Yes

Reviewer #2: Yes

5. Review Comments to the Author

Reviewer #1: Firstly, this article lack novelty. Vaccine safety and adverse reactions had been published for years and in many different aspects. This study use trivalent influenza vaccine as research target, which showed no originality. Secondly, the method of administrating the vaccine is too roughly written. It should include the route of administration, cell type of the vaccine (cell-derived or egg-derived). Third, some of the subgroups' patient amount is small which make readers concern if the results are promising or not. Last but not least, the percentages of adverse events between subgroups are wide and significant which raise concern on the credibility of the data.

Reviewer #2: The manuscript describes active pharmacovigilance of seasonal flu vaccine in Brazil. The study was well conducted with follow-up at appropriate time points. Relevant results are reported although the manuscript would benefit from clarity in terms of linking the specific objectives to the results.

Content:

-The authors state the “all local adverse events were considered as adverse reactions to IB influenza vaccine”, which is a reasonable assumption. However in the tables the systemic events were also considered to be adverse reactions, which is more problematic as there may be other reasonable causes. Furthermore, in the objective the authors state they look for AEFIs but in the results nearly everything is called a reaction.

-It is not always clear which timeframe (14 days, 42 days) is considered, this information should be added to each section in the results. The secondary outcomes included unsolicited AEFIs at two timepoints but only one set of results is reported (without specified timeframe).

-The primary outcome is ‘incidence/causality/duration/seriousness’ of AEFIs in 42 days. The primary outcome appears to be described in the section ‘adverse events and adverse reactions’ however this paragraph mentions only incidence/causality and not duration/seriousness.

-The secondary outcomes included unsolicited AEs during 14 days and during 42 days. This distinction does not appear to be present in the results.

-The statement that at 42 days no neurological signs/ symptoms associated with GBS were reported should (also) be included in the results, to answer one of the exploratory objectives.

The exploratory objective “frequency of AEFI according to WHO causality assessment classification”. Is this the classification of an event as a reaction? Should be more clearly framed in the results.

Tables:

-General: to add the timeframe considered to the table title

-Table 2

• footnotes are missing

• to add % of participants with AE

Textual considerations:

-Check consistent use of abbreviations (e.g. TIV in section study design)

-Introduction

• Last paragraph, first sentence: delete ‘influenza’

-Subjects

• “documented with the signature of the informed consent for by THE subject or A legal representative (…) temperature over 37.8 oC on THE vaccination day (…) affects the participant’S ability”

• Primary, secondary and exploratory endpoints

• There is a lonely bracket in the first sentence

-Studies Procedures for Active Surveillance

• To replace volunteer by participant

• Not clear what the “+2 days” stands for

• “Before LEAVING, (…) THEY received (…) check the baby’S health status”

• “The child was followed up to ASSESS averse event causality”

• There are two commas after reference 9

-Statistical analysis

• Delete ‘its’ in the second sentence

-Participants

• “most participants WERE female”

• It is confusing to refer to non-enrolled persons as ‘participants’ because they didn’t actually participate in the study

6. PLOS authors have the option to publish the peer review history of their article (what does this mean?). If published, this will include your full peer review and any attached files.

Reviewer #1: No

Reviewer #2: No

---

## [Author Response · Author response to Decision Letter 0]

28 Dec 2020

Dear Sir/Madam,

We are grateful for PLOS ONE Reviewer’s comments on our manuscript entitled “Active pharmacovigilance of the seasonal trivalent influenza vaccine produced by Instituto Butantan: a prospective cohort study of five target groups”, PONE-D-20-21550, submited for publication in PLOS One.

We have carefully modified the manuscript and supplement in order to address all reviewers’ recommendations. Please find below our answer to points raised by the reviewers: 

Journal Requirements:

1. We have carefully read the web instructions provided, so that the manuscript meets PLOS ONE's style requirements, including those for file naming; 

Authors: Instructions were carefully read and the manuscript modified accordingly. 

2. Please provide the manufacturer information and lot numbers of the vaccines:

Authors: The lot numbers and manufacturer information of the vaccines administered in both studies (2017 and 2018) are listed below and included in the Material and Methods – Study Vaccines (page 10, lines 210, 212 and 213): 

Secretaria de Estado da Saúde – Instituto Butantan

FLU-05-IB – 2017:

Centro Buritis: Lot IB 170047

Centro CRIE Unifesp: Lot IB 170074

FLU-06-IB - 2018

Centro CRIE Unifesp: Lot IB 180069 and Lot IB 180076

The respective package insert of 2017 and 2018 products are enclosed.

3. Thank you for stating in the text of your manuscript that you obtained a "signature of the Informed Consent Form by subject or legal representative". Please also add this information to your ethics statement in the online submission form. In addition, please clarify whether parents/guardians gave consent for minors.

Authors: As recommended, we have added the requested information in the ethics statement. Regarding the consent procedures, the team of each research centre obtained a signature of the Informed Consent Form from parents or the child’s legal representative, when applicable. 

4. "TV, MGS, JYKV, RPG, MTRPC, ROP, PEB, AS, HME, JPS, VLG, MEK, MBBL, MMMO, SHDNR, LYW and ARP are employees of Instituto Butantan. BCT and GM are former employees of Instituto Butantan. SHDNR and LYW received financial aid for their roles as principal investigators." Please confirm that this does not alter your adherence to all PLOS ONE policies on sharing data and materials, by including the following statement: "This does not alter our adherence to PLOS ONE policies on sharing data and materials.” (as detailed online in our guide for authors http://journals.plos.org/plosone/s/competing-interests). If there are restrictions on sharing of data and/or materials, please state these. Please note that we cannot proceed with consideration of your article until this information has been declared.

Authors: We have read the PLOS ONE guidance for authors on competing interests. The abovementioned competing interests do not alter our adherence to all PLOS ONE policies on sharing data and materials. There are no restrictions on sharing data and materials. We have included the statement "This does not alter our adherence to PLOS ONE policies on sharing data and materials.”, page 25, line 463.

Reviewers’ Comments to the Authors

5. Reviewer #1: Firstly, this article lack novelty. Vaccine safety and adverse reactions had been published for years and in many different aspects. This study uses trivalent influenza vaccine as research target, which showed no originality. Secondly, the method of administrating the vaccine is too roughly written. It should include the route of administration, cell type of the vaccine (cell-derived or egg-derived). Third, some of the subgroups' patient amount is small which make readers concern if the results are promising or not. Last but not least, the percentages of adverse events between subgroups are wide and significant which raise concern on the credibility of the data.

Authors: In times of decreasing vaccine acceptance due to safety concerns, it has become increasingly relevant to publish rigorously conducted pharmacovigilance studies. These two prospective cohorts studies were able to provide original information on the safety of Instituto Butantan trivalent influenza for five target-groups, including pregnant and post-partum women. As requested, we have included additional information regarding the route of administration (IM) and cell type of vaccine (egg-derived) in the first paragraph of Material and Methods (page 6, lines 106-108). The study followed European Medicines Agency’s guidelines in order to define the sample size of each subgroup. As pointed out the reviewer, the comparison of AR frequency among subgroups is limited by different characteristics and sample sizes, as we included in the discussion (page 22, lines 377-379) 

6. Reviewer #2: The manuscript describes active pharmacovigilance of seasonal flu vaccine in Brazil. The study was well conducted with follow-up at appropriate time points. Relevant results are reported although the manuscript would benefit from clarity in terms of linking the specific objectives to the results.

Authors: Thank you for your comments, we have improved the text in order to better link the objectives to the results, which can be seen in page 13, line 260 and 268, page 14, line 276, page 15, line 289, page 16, line 302, page 17, line 315, page 18, line 328, page 20, line 342, page 21, lines 354, 361 and 362.

Content comments

7. The authors state the “all local adverse events were considered as adverse reactions to IB influenza vaccine”, which is a reasonable assumption. However in the tables the systemic events were also considered to be adverse reactions, which is more problematic as there may be other reasonable causes. Furthermore, in the objective the authors state they look for AEFIs but in the results nearly everything is called a reaction

Authors: Thank you for your comments. Regarding the systemic adverse reactions, they were all defined after a causality assessment using “Uppsala Monitoring Centre” (UMC) of the World Health Organization (WHO) criteria to every Adverse Event recorded as detailed in page 10, lines 197-199. In order to further clarify, we have included the following statement “Adverse Reaction (AR) was defined as any AE with a reasonably causal relationship with the vaccine, according to UMC-WHO” below table 2-8.

8. It is not always clear which timeframe (14 days, 42 days) is considered, this information should be added to each section in the results. The secondary outcomes included unsolicited AEFIs at two timepoints but only one set of results is reported (without specified timeframe).

Authors: Thank you for your comments. The specific timeframe was added to each section in the respective tables, as can be seen in tables 2 to 8.

9. The primary outcome is ‘incidence/causality/duration/seriousness’ of AEFIs in 42 days. The primary outcome appears to be described in the section ‘adverse events and adverse reactions’ however this paragraph mentions only incidence/causality and not duration/seriousness.

Authors: Thank you for your comments. We have modified the paragraph to include additional information regarding duration/seriousness in page 13, lines 266-267. Additional tables with more detailed information on duration/seriousness are provided as supplementary material 2. 

10. The secondary outcomes included unsolicited AEs during 14 days and during 42 days. This distinction does not appear to be present in the results.

Authors: Thank you for your comments. We have included this information in detail in the supplementary material 2.

Tables comments

11. General: to add the timeframe considered to the table titles.

Authors: Duration period referred to data in each table was added in all table legends.

12. Footnotes in table 2 is missing.

Authors: Footnote in table 2 was added.

13. Add % of participants with AE.

Authors: We have included the percentage of participants with AE and AR in table 2.

Textual considerations

14. Check consistent use of abbreviations (e.g. TIV in section study design).

Authors: All abbreviations were checked in the section Study Design and in the whole text.

15. Introduction: Last paragraph, first sentence: delete ‘influenza’.

Authors: The word “influenza” was deleted from the first paragraph of introduction.

16. Subjects: “documented with the signature of the informed consent for by THE subject or A legal representative (…) temperature over 37.8 oC on THE vaccination day (…) affects the participant’S ability”. 

Authors: All the capital letter words suggested were inserted the text, page 7, lines 128, 131 and 135.

17. Primary, secondary and exploratory endpoints: There is a lonely bracket in the first sentence.

Authors: The lonely bracket was removed from the sentence.

18. Studies Procedures for Active Surveillance: To replace volunteer by participant.

Authors: The word volunteer was replaced by participant in the abovementioned text in page 8.

19. Studies Procedures for Active Surveillance: Not clear what the “+2 days” stands for.

Authors: We add a clarification about the “+2 days” and the sentence was reframed as follows: “…. Before leaving, participants were instructed to return to the study site for a visit on Day 3 considering 2 extra days tolerance (+2 days), answer a telephone call on Day 7 (+3 days) post-vaccination and return for the final visit on Day 14 (+4 days) post-vaccination, they received the Participant’s Diary to be filled with both solicited and unsolicited AEFI occurred within the 14 days post- vaccination and they were instructed to inform the study staff immediately whenever there was hospitalization and/or serious adverse events…” in page 8, lines 158-164.

20. Studies Procedures for Active Surveillance: “Before LEAVING, (…) THEY received (…) check the baby’S health status”.

Authors: All recommendations were included in the text, in page 8, lines 158-164.

21. Studies Procedures for Active Surveillance: “The child was followed up to ASSESS averse event causality”

Authors: The sentence “… the child was followed up to determination of the adverse event causality” was reframed as suggested as follows:“…the child was followed up to assess adverse event causality”, page 9, lines 174-175.

22. Studies Procedures for Active Surveillance: There are two commas after reference 9.

Authors: Extra comma was removed from the text. 

Statistical Analysis considerations

23. Statistical Analysis: Delete “its” in the second sentence.

Authors: The word was deleted from the second sentence. 

Results considerations

24. Participants: “Most participants WERE female”

Authors: The singular verbal tense was corrected to plural, in page 12, line 254.

25. Participants: It is confusing to refer to non-enrolled persons as ‘participants’ because they didn’t actually participate in the study

Authors: The sentence was reframed to improve understanding as follows: “…The total number of invited participants from the 2017 and 2018 studies was 949 (546 and 403 from the 2017 and 2018 studies, respectively) (Fig 1). Seven (7) out of 949 invited participants were excluded from the study (4 unavailable for the entire study period, 1 no indication for vaccination, and 2 unknown reasons). Among the 942 enrolled participants, 305 were elderly, 109 children, 108 pregnant women, 32 postpartum women, and 388 healthcare workers…”, page 12, line 246-250.

We hope to have properly addressed the points raised by the editors and reviewers. Thank you for the opportunity to improve the manuscript. Please do not hesitate, in case additional modifications are needed. 

Yours truly,

Tazio Vanni, MD PhD

(Corresponding Author)

Pharmacoepidemiology and Pharmacovigilance Research Manager

Clinical Safety and Risk Management Center 

Instituto Butantan

---

## [Decision Letter · Decision Letter 1]

21 Jan 2021

Active pharmacovigilance of the seasonal trivalent influenza vaccine produced by Instituto Butantan: a prospective cohort study of five target groups

PONE-D-20-21550R1

Dear Dr. Vanni,

We’re pleased to inform you that your manuscript has been judged scientifically suitable for publication and will be formally accepted for publication once it meets all outstanding technical requirements.

Kind regards,

Ray Borrow, Ph.D., FRCPath

Academic Editor

PLOS ONE

Additional Editor Comments (optional):

Reviewers' comments:

Reviewer's Responses to Questions

**Comments to the Author**

1. If the authors have adequately addressed your comments raised in a previous round of review and you feel that this manuscript is now acceptable for publication, you may indicate that here to bypass the “Comments to the Author” section, enter your conflict of interest statement in the “Confidential to Editor” section, and submit your "Accept" recommendation.

Reviewer #2: All comments have been addressed

2. Is the manuscript technically sound, and do the data support the conclusions?

Reviewer #2: (No Response)

3. Has the statistical analysis been performed appropriately and rigorously? 

Reviewer #2: (No Response)

4. Have the authors made all data underlying the findings in their manuscript fully available?

Reviewer #2: (No Response)

5. Is the manuscript presented in an intelligible fashion and written in standard English?

Reviewer #2: (No Response)

6. Review Comments to the Author

Reviewer #2: (No Response)

7. PLOS authors have the option to publish the peer review history of their article (what does this mean?). If published, this will include your full peer review and any attached files.

Reviewer #2: No

---

## [Editor Report · Acceptance letter]

25 Jan 2021

PONE-D-20-21550R1 

Active pharmacovigilance of the seasonal trivalent influenza vaccine produced by Instituto Butantan: a prospective cohort study of five target groups 

Dear Dr. Vanni:

I'm pleased to inform you that your manuscript has been deemed suitable for publication in PLOS ONE. Congratulations! Your manuscript is now with our production department. 

Kind regards, 

on behalf of

Prof. Ray Borrow 

Academic Editor

PLOS ONE